# Enhancing Hand Sensorimotor Function in Individuals with Cervical Spinal Cord Injury: A Novel Tactile Discrimination Feedback Approach Using a Multiple-Baseline Design

**DOI:** 10.3390/brainsci15040352

**Published:** 2025-03-28

**Authors:** Ken Kitai, Kaichi Nishigaya, Yasuhisa Mizomoto, Hiroki Ito, Ryosuke Yamauchi, Osamu Katayama, Kiichiro Morita, Shin Murata, Takayuki Kodama

**Affiliations:** 1Rehabilitation Department, Maizuru Red Cross Hospital, Maizuru City 624-0906, Kyoto, Japan; 2Graduate School of Health Sciences, Kyoto Tachibana University, Kyoto City 607-8175, Kyoto, Japan; h901524007@st.tachibana-u.ac.jp (H.I.); h901522007@st.tachibana-u.ac.jp (R.Y.); murata-s@tachibana-u.ac.jp (S.M.); kodama-t@tachibana-u.ac.jp (T.K.); 3Rehabilitation Department, Zenjokai Rehabilitation Hospital, Nagoya City 457-0046, Aichi, Japan; k.nishigaya0913@gmail.com; 4Rehabilitation Department, Watanabe Hospital, Chita 470-3235, Aichi, Japan; n.y.m0809@gmail.com; 5Department of Preventive Gerontology, Center for Gerontology and Social Science, Research Institute, National Center for Geriatrics and Gerontology, Obu City 474-8511, Aichi, Japan; katayama.o@ncgg.go.jp; 6Cognitive and Molecular Research Institute of Brain Diseases, Kurume University, Kurume 830-0011, Fukuoka, Japan; kiichiro@kurume-u.ac.jp

**Keywords:** cervical spinal cord injury, hand sensorimotor dysfunction, tactile discrimination compensatory real-time feedback device, multiple-baseline design

## Abstract

**Background/Objectives**: This study evaluated the effects of a tactile-discrimination compensatory real-time feedback device on hand sensorimotor function in cervical spinal cord injury patients. The study assessed changes in hand numbness, dexterity, and electroencephalogram (EEG) activity, particularly γ-wave power in the sensorimotor area during skilled finger movements. **Methods**: Three patients with cervical spinal cord injury who presented with hand sensorimotor dysfunction underwent treatment with this device. All cases underwent the intervention using an AB design; A is the exercise task without the system device, and B is the exercise task under the system device. To confirm the reproducibility and minimize the influence of confounding factors, a multiple-baseline design, in which the intervention period was staggered for each subject, was applied. To determine efficacy, the hand numbness numerical rating scale, peg test, and EEG were measured daily, and Tau-U calculations were performed. **Results**: In two of three cases, moderate or very large changes were observed in numbness in B. In all cases, there was a large or very large change in the peg test results in the B. Regarding EEG activity, the non-skilled participants showed amplification of γ-wave power in the sensorimotor area during the B. Conversely, in the skilled participants, the γ-wave power of the sensorimotor area was attenuated during skillful movements. **Conclusions**: These findings indicate that the ability of the brain to compare and align predictive control with sensory feedback might be compromised in patients with damage to the afferent pathways of the central nervous system. Moreover, the use of this device appears to have played a role in supporting functional recovery.

## 1. Introduction

Cervical spinal cord injury (SCI) is a general term for diseases caused by cervical spinal cord damage due to factors such as trauma due to traffic accidents [1] or age-related cervical degeneration [2]. This condition can occur in people of all ages, impair motor function, and significantly reduce the ability to perform daily activities. The lifetime cost of care and rehabilitation is estimated to exceed one million United States dollars per patient [1]. Restoration of motor function in patients with cervical SCI is an urgent issue to reduce the burden on patients, their families, and society.

In severe cases of nerve compression, cervical SCI is often treated surgically to relieve compression [3]. However, patients with mild-to-moderate cervical SCI with a modified Japanese Orthopedic Association score of 12–16 reflecting neurological symptoms in the upper and lower extremities were followed for 10 years. The results showed no significant difference in activities of daily living between conservative treatment, which did not relieve spinal cord compression, and surgical treatment, which did relieve spinal cord compression [3]. One narrative review reported that, in cases of incomplete cervical SCI, neurological symptoms may improve if surgical treatment is performed to reduce nerve compression [4]. However, this narrative review also reports that surgical treatment may cause further damage to the spinal cord and that nerve recovery may be better promoted if treatment is conservative and promotes natural healing. Therefore, the superiority of conservative or surgical treatments remains debatable, and it is believed that surgical and other treatment methods should be used in combination.

Rehabilitation is a nonsurgical treatment for cervical SCI. As the nerves supplying motor and sensory control to the upper limbs exit from the cervical spinal cord, the motor function of the upper limbs and fingers is easily impaired [5,6]. Rehabilitation to restore this function is considered the highest priority [7]. Walker and Detloff [7] suggested that activity dependence through high-frequency exercise and rehabilitation utilizing sensory input can promote neuroplasticity in damaged nerves exiting from the cervical spinal cord and improve motor function. Considering this background, interventions using sensory input associated with movements, such as trans-spinal stimulation and transcutaneous simulation, have recently been used to improve upper limb motor function, and it has been reported that this significantly improves upper limb motor function and muscle output of the target muscles [8].

Serious and frequent sequelae that reduce motor function in the upper limbs after cervical SCI include motor paralysis and finger numbness [6]. In fact, studies show that numbness affects over 80% of individuals with cervical SCI [9]. This sensory impairment severely limits fine motor control because the finger pads detect the friction information generated when touching an object and adjust the muscles of the fingers accordingly, allowing for dexterous movements such as object manipulation that are necessary for daily living activities [10,11]. The ability to sense the frictional information is impaired, and finger muscle activity is unable to accurately respond when numbness occurs, leading to sensorimotor dysfunction and functional limitations [12,13]. While spontaneous recovery of dexterity in patients with cervical SCI is difficult due to the limited regenerative capacity of axons in the central nervous system, rehabilitation can promote dexterity through exercises using original movement patterns [14,15]. However, interventions such as trans-spinal and transcutaneous stimulation, while beneficial for gross motor function, lack the specificity of feedback required for restoring fine finger control, as they cannot provide electrical stimulation in response to real-time frictional information generated during object contact.

To address this issue, Kitai et al. [16] attempted to use a system to provide intervention for patients with sensorimotor dysfunction of the fingers following cervical SCI. In that system, frictional information generated by touching the finger is frequency-modulated and fed back in compensatory form to other body parts as vibration information. Consequently, interventions using this device have been reported to reactivate the sensorimotor region of the cortex during dexterous movements. This suggests that this may reflect the initial process of motor learning during dexterous movements. Kitai et al. [17] have previously used a similar system of intervention in patients with cervical SCI. From the second week of intervention, they observed an attenuation of the coherence values of the median frontal-median parietal theta wave (4–8 Hz), which is thought to be involved in cognitive processing during movement. They report that this may reflect a decrease in cognitive load during fine motor movements (reflecting a later stage of motor learning) compared to pre-intervention. Interventions using the system described above suggest that it may be an appropriate feedback stimulus for skillful movement in individuals with hand sensorimotor dysfunction after cervical SCI. On the other hand, in testing the effectiveness of interventions using this system, no study has been conducted on the indications for differing types of cervical SCI. In recent years, there has been growing interest in the effectiveness of treatment based on the severity of sensorimotor dysfunction of the fingers after cervical SCI. Kalli–Ryan et al. [18] investigated the effectiveness of treatments for finger sensorimotor dysfunction after cervical SCI based on severity. However, the optimal treatment method based on severity has not been clarified. Thus, the selection of an appropriate treatment method based on the severity of sensorimotor dysfunction is considered a future research topic because the high-frequency band of the electroencephalogram (EEG) in the sensorimotor area is believed to enable a highly accurate estimation of finger movements, regardless of the severity of motor impairment [19]. Therefore, to determine the suitability of this device based on the severity of the condition, verifying the intervention effect using high-frequency EEG bands rather than EEG imaging analysis or low-frequency EEG bands would be optimal.

The aim of this study was to intervene with a tactile-discrimination compensatory real-time feedback device in patients with cervical SCI who present with hand sensory-motor dysfunction of various severities.

## 2. Materials and Methods

### 2.1. Case Introduction

The inclusion criteria for this study were as follows: Mini-Mental State Examination score of 27 or higher, American Spinal Injury Association (ASIA) score criteria reference, C6–8 motor function of 2 or higher, the finger numbness NRS of 1 or higher, and who do not have higher brain dysfunction. The exclusion criteria for this study were as follows: Mini-Mental State Examination score of 26 or less [20], referring to the American Spinal Injury Association (ASIA) score criteria, individuals with C6–8 motor function of 1, those with 0 on the finger numbness numerical rating scale (NRS), inability to consent to participate in this study, and diagnosis with higher brain dysfunction. The subjects were three patients with cervical SCI (Table 1). Interventions and measurements were performed for Case I at Hospital A and for Cases II and III at Hospital B (a two-institution joint study). Case I is an 83-year-old woman. Case I became aware of decreased ability to perform skillful movement of the right fingers 2.5 months before the start of the study; cervical spondylotic myelopathy was diagnosed via Magnetic Resonance Imaging (MRI). Case I underwent a cervical laminectomy (C3/4) 2 months prior to study entry. Case II is a 51-year-old man. Case II had been aware of decreased skillful movement in his right fingers for 3 years prior to the start of the study. Case II was diagnosed with cervical spondylotic myelopathy (C3–6) by MRI two months before the study began. Case II underwent cervical laminectomy (C3–6) two weeks before the study began. Case III is a 53-year-old man. Three months before the study began, Case III fell 1.5 m on his bicycle and was diagnosed with cervical SCI (C7) by MRI. The right hand was affected in all cases. In Case I, the ASIA score for C6–C8 motor function was 3 and for sensory function was 2; the finger numbness NRS score was 1 [16,17,21], while her Purdue Pegboard Test (peg test) score [22], a known index of dexterity ability, averaged 6 after two sessions. In Case II, for the ASIA scores, C6–7 was 2 and C8 was 3 for motor function; for sensory function, C6–8 was 1. finger numbness NRS score was 8, and the average score after the two peg tests was 1.5. In Case III, for the ASIA scores, C6 was 4 for motor function, C7 was 5, and C8 was 4. For sensory function, C6–8 was 1. The finger numbness NRS score was 4, and the average score after the two-peg tests was 10. This study aimed to improve the patient’s numbness and dexterity. This study was approved by the Kyoto Tachibana University Ethics Committee (approval number 23–27). The purpose, content, and procedures of the study were explained to the participants both verbally and in writing, and informed consent was obtained.

### 2.2. Intervention Methods

In dexterous movements, the friction information generated when the finger pad touches an object is detected in the brain, and the corresponding finger muscle activity occurs [11]. To reproduce these specific finger functions, a sensor that can detect the friction information generated when a finger pad touches an object is required. Therefore, a system called a finger recorder (hereinafter referred to as “Yubireco”) (Figure 1) was developed (Tekkigihan Co., Ltd., Kyoto, Japan) that detects the friction information generated when a finger pad touches an object with a sensor and feeds the friction information back to other body parts in real-time as a vibration stimulus [23] and used in this study. The vibration sensor was wrapped around the distal interphalangeal joint of the index finger, and the output from the sensor was modulated to a frequency that humans could sense, allowing the vibration information to be presented through a vibrator in the bi-temporal bone, making it possible to synchronously match the visual information accompanying finger movements and provide real-time feedback of the vibration information accompanying finger movements.

Regarding the research protocol, for intervention verification using real-time feedback of tactile discrimination compensation for finger sensorimotor dysfunction after cervical SCI, the effectiveness of the intervention was verified using an AB design [17]. Therefore, the intervention was implemented for all participants using an AB design, where A was the baseline period, and B was the intervention period. In the A, the motor tasks were performed. The motor tasks were based on the method described previously [16,17,21] and consisted of a 10 min task of stacking square building blocks used in the box and block test, a 10 min task of discriminating the roughness of five pieces of sandpaper using the ventral part of the index finger, and a 10 min task of inserting a 25 mm long and 3 mm wide iron peg (Sakai Medical Co., Ltd., Tokyo, Japan) into a board with two vertical rows of 25 holes (peg task). In the B, the participants were fitted with a Yubireco, and the same motor tasks were performed. Further, a multiple baseline design (MBD), a method for shifting the intervention period for each participant, was used [24]. The verification of intervention effects using MBD has attracted attention in recent years. Using MBD, it is possible to confirm the reproducibility of the intervention effects while minimizing the influence of confounding factors. Three or more data points are required for A and B as criteria for performing MBD. In addition, three or more subjects were required to increase the internal validity by confirming reproducibility three or more times. Furthermore, to eliminate the influence of natural recovery and improvement from the intervention performed during the baseline period of Phase A, processing is recommended to correct the trend from the baseline [25]. Taking these factors into consideration, we decided to carry out an intervention using MBD for three patients with cervical SCI, with a total intervention period of 14 days (AB) and processed the data by considering the baseline trend (Figure 2).

Capable of feeding back tactile sensory information to the bilateral temporal bones as vibration information when touched with fingers via a transducer.

All cases were assigned to the protocol shown in the figure, and the study was conducted.

The evaluation included a longitudinal evaluation and pre- and post-assessments. Longitudinal evaluation included daily neurophysiological, motor, and sensory function assessments. Pre- and post-assessments were performed before A started, after A ended (before B), and after B ended. Learning was evaluated as a part of the assessment.

For longitudinal and neurophysiological evaluations, EEG activity was measured daily for 1 min during the pegboard task after the motor task. EEG measurements were performed in a quiet environment with participants in a relaxed state. EEG measurements were performed once without wearing the Yubireco and using a portable electroencephalograph StEEG (Creact Co., Ltd., Tokyo, Japan). EEG measurements were performed at eight sites based on the international 10–20 system: Fp1, Fp2, T7, T8, O1, O2, Fz, and Pz, with both earlobes used as reference electrodes. The sampling frequency was set at 1000 Hz. In the motor function evaluation, the peg test [22] was conducted to evaluate finger dexterity. The peg test is an examination in which participants insert as many iron pins as possible, each 25 mm long and 3 mm wide, within 30 s into a board (Sakai Medical Co., Ltd., Tokyo, Japan), with 25 holes arranged vertically into two rows. The measurements were performed twice daily after the motor task. For sensory function evaluation, the numbness NRS was administered once daily after the motor task.

The Motor Activity Log-14 (MAL-14), which evaluates the amount and quality of finger use, was used for pre- and post-learning evaluations. The evaluation of upper limb use using MAL is widely used to evaluate the effectiveness of rehabilitation for upper limb motor dysfunction after stroke [26]. In this study, the MAL-14 was used to evaluate the use of the upper limbs in real life using 14 items. The evaluation method included the amount of use of the upper limbs (AoU) (0 = not used at all and 5 = used as much as before the injury) and quality of movement (QoM) (0 = cannot move at all and 5 = can move as well as before). In addition, finger movement sense of agency NRS [16,17,21] was obtained for each item in the MAL-14, and the relationship between behavioral change and sense of agency was evaluated. Sense of agency is defined as the feeling that one is in control of one’s own movements. The sense of agency (SoA) was assessed by asking, “To what extent do you feel that you are the one performing your own exercise?” on the NRS (0 = not at all; 10 = extremely strong).

### 2.3. Analysis Method

For longitudinal evaluation, no band-pass filtering was performed in this case to extract a wide range of frequencies (40–100 Hz). We performed independent component analysis (ICA) to remove noise components as a preprocessing step for the EEG data. ICA is a statistical technique that separates a multivariate signal into independent subcomponents, assuming that the subcomponents are non-Gaussian signals and are statistically independent of each other [27]. In the context of EEG data, ICA helps to identify and separate various sources of neural activity as well as artifacts such as eye blinks, muscle activity, and cardiac signals. The independent components extracted by ICA represent distinct sources of neural activity or artifacts that contribute to the overall EEG signal rather than conventional EEG parameters like average amplitude or power of spectral peaks. To perform ICA, we used EEGLAB, a toolbox in MATLAB (Matrix Laboratory) (version MATLAB R2023b) [28]. We used the Infomax algorithm implemented in EEGLAB [29] with the default parameters (learning rate: 0.001, batch size: 1, maximum number of iterations: 512). The independent components were extracted from the EEG data using ICA. From the components separated by ICA, the premixed signal was estimated under the assumption that it was a linear sum of statistically independent components, and clean EEG data were extracted by removing the components representing noise and artifacts. The noise-processed data were subsequently log-transformed using a custom MATLAB script [30]. By performing a log transformation, the weak currents in the EEG are raised to a measurable range. After normalization, the power value (decibel: dB) of the frequency band of interest, which indicates the abundance of frequency components in the region of interest, was calculated from the cleaned EEG data. These values were calculated using methodology, e.g., Welch’s method or multitaper spectral estimation, which estimates power spectral density (PSD) for specific frequency bands. The units for these measurements are microvolts squared per Hertz (μV^2^/Hz), which is the standard unit for quantifying EEG power spectral density. The format represents the mean power across trials or participants and its variability (standard deviation).

The region of interest was between the Fz and Pz gamma wavebands (40–100 Hz). Although the measurement sites for Fz and Pz are different, the same frequency band was used, and the same amplification settings were used. The Fz and Pz gamma wave power values obtained in each period were summed, and the average value and standard error were calculated. These values were compared between periods. The values obtained in each period of the peg test and Numbness NRS scores were subjected to Tau-U calculations to obtain the effect size, which is a statistical index that indicates the magnitude of the effect. Tau-U is characterized by its ability to combine non-overlapping periods and trends in the intervention phase and correct baseline trends [31]. This allowed the verification of the effect of Intervention B in light of the effects of natural recovery and Intervention A. The effectiveness of the results obtained using Tau-U was judged as “no change or small change” when Tau-U was 0–0.20, “moderate change” when it was 0.20–0.60, “large change” when it was 0.60–0.80, and “very large change” when it was 0.80–1 [32]. Tau-U analysis was performed using the web application software Single Case Study Web-Based Calculator for SCR Analysis (version 2.0) [33]. The software then combined the A and B values obtained in each of the three cases to obtain the overall effect size (Tau-U). For the pre- and post-assessments, the AOU and QOM scores of the MAL-14 were calculated by dividing the total score (0–5) by the number of items. The sense of agency (NRS) scores were calculated by dividing the total score (0–10) by the number of items. These were compared between periods.

## 3. Results

Comparing the EEG results between A and B, the Fzγ wave in Case I was amplified in B from 0.015 ± 0.008 to 0.016 ± 0.019 (6.7% increase), while the Pzγ wave was amplified from 0.008 ± 0.001 to 0.014 ± 0.013 (75.0% increase). The Fzγ wave in Case II was amplified from 0.074 ± 0.038 to 0.090 ± 0.036 (21.6% increase), while the Pzγ wave was amplified from 0.007 ± 0.003 to 0.055 ± 0.023 (685.7% increase). The Fzγ wave in Case III was attenuated from 0.744 ± 0.281 to 0.010 ± 0.003 (98.7% decrease), while the Pzγ wave was attenuated from 0.852 ± 0.322 to 0.033 ± 0.017 (96.1% decrease) (Table 2).

Regarding the peg test, Case I showed a large change from 6.50 ± 0.35 to 7.30 ± 0.26 (12.3% increase). Case II showed a large change from 2.30 ± 0.34 to 3.28 ± 0.30 (42.6% increase). Case III shows a very large change from 11.90 ± 0.38 to 14.90 ± 0.26 (25.2% increase). All patients showed a large or very large change in the Tau-U scores. Tau-U for Combined cases I–III is 0.80 (Table 3).

Regarding finger numbness NRS scores, Case I showed no change from 1.00 ± 0.00. Case II exhibited a moderate change from 7.80 ± 0.20 to 7.11 ± 0.31 (8.8% decrease). Case III showed a very large change from 3.70 ± 0.49 to 1.00 ± 0.58 (73.0% decrease). Two of the three patients showed moderate or very large changes in the Tau-U scores. Tau-U for Combined cases I–III is 0.65 (Table 4).

Regarding MAL, the AoU scores of Case I were 2.85 points before Intervention A, 3.08 points after Intervention A, and 3.62 points (17.5% increase) after Intervention B. The QoM scores were 2.46 points before Intervention A, 2.54 points after Intervention A, and 3.23 points (27.2% increase) after Intervention B. The SoA scores were 5.69 points before Intervention A, 5.69 points after Intervention A, and 6.92 points (21.6% increase) after Intervention B. The AoU scores of Case II were 0.64 points before Intervention A, 1.14 points after Intervention A, and 1.50 points (31.6% increase) after interventions A and B, respectively. The QoM scores were 0.43 points before Intervention A, 1.36 points after Intervention A, and 1.43 points (5.1% increase) after Intervention B. After Intervention A, the SoA scores were 0.79 points before Intervention A, 1.29 points after Intervention A, and 1.64 points (27.1% increase) after Intervention B. The AoU scores of Case III were 4.07 points before Intervention A, 4.21 points after Intervention A, and 4.71 points (11.9% increase) after Intervention B. QoM scores were 4.07 points before Intervention A, 4.21 points after Intervention A, and 4.71 points (11.9% increase) after Intervention B. SoA scores were 8.29 points before Intervention A, 8.71 points after Intervention A, and 9.43 points (8.3% increase) after Intervention B (Table 5).

All patients showed a large or very large change in the Tau-U scores. Tau-U scores for Combined cases I–III are a very large change.

Two of the three patients showed moderate or very large changes in the Tau-U scores. Tau-U scores for Combined cases I–III is a large change.

All patients showed a tendency for improvement in B.

## 4. Discussion

These results suggest that an intervention using a real-time feedback device for tactile discrimination compensation may improve sensorimotor function in patients with sensorimotor dysfunction of the fingers following cervical SCI. In particular, the improvement in sensorimotor function was found to be greater in moderate-to-severe cases, suggesting that interventions using this device may be more effective in such cases.

The three participants in this study all experienced numbness in their fingers and a decline in dexterity; therefore, they underwent rehabilitation for finger sensorimotor dysfunction. However, no study has yet used the MBD to examine the effects of rehabilitation on finger sensorimotor dysfunction after cervical SCI. We believe that using the MBD to verify the effect of rehabilitation on finger sensorimotor dysfunction after cervical SCI may enable a more detailed examination of the reproducibility of the intervention effect and the influence of confounding factors. Therefore, the effectiveness of the rehabilitation was evaluated using high-frequency EEG bands with an MBD design for patients with different levels of finger sensorimotor dysfunction.

In this study, Fz-Pz was used as the region of interest in order to examine whether or not there is an effect on sensory-motor function. The evaluation of sensorimotor function using EEG has attracted considerable attention in recent years. Quandt et al. [34] previously analyzed the EEG activity during dexterous finger movements in older participants. It has been reported that the EEG activity in the sensorimotor region of the cortex changes with a decline in finger dexterity. Thus, changes in sensorimotor function can be objectively evaluated using EEG. Changes in sensorimotor function can further be evaluated in greater detail by verifying the effects of rehabilitation using EEG, even in cases of finger sensorimotor dysfunction after cervical SCI. Preparatory neural activity was detected in the supplementary motor cortex (via Fz), which is crucial for early-stage motor planning. Monitor parietal lobe engagement (via Pz), which supports sensory integration during task execution [35,36]. Our analysis focused on network-level dynamics relevant to sensorimotor function. We employed gamma waves in the sensorimotor area because the strength of gamma waves correlates with the severity of upper limb motor dysfunction and because gamma waves can accurately assess the quality of hand movement regardless of severity [19,37]. It has been reported that gamma waves in the sensorimotor area are amplified during actual movements and are related to perception, attention, and motor control during movement [38,39]. Sensorimotor gamma waves are believed to integrate sensory and motor processes during motor control and promote adaptive motor control during voluntary movements [40,41]. These findings indicate that gamma waves in the sensorimotor area function when the integration of sensory and motor information necessary for motor control is enhanced.

Regarding motor function evaluation, the peg test results showed a large change in Tau-U in Cases I and II, and the gamma wave power values in Fz and Pz were amplified. Regarding the sensory function evaluation, Case II showed moderate changes in Tau-U in the numbness NRS, and Case III showed very large changes in Phase B. Sensorimotor gamma waves are said to function when sensorimotor adaptive control during voluntary movement is required and integrate sensory and motor processes during motor control [40,41]. Sensorimotor adaptive control includes the readjustment of the brain mapping between motor commands and sensory feedback in response to motor errors, and it has been reported that when brain mapping is readjusted, the gamma power value in the sensorimotor region is amplified [42]. In Cases I and II, gamma wave power in the sensorimotor area increased during the intervention period using Yubireco, suggesting that Yubireco may improve the sensorimotor adaptive control ability required for dexterous movements. These results are in accordance with those of Gomes-Osman and Field-Fote [43]. This finding supports the possibility that an intervention combining sensory feedback and sensorimotor training may be effective in improving finger sensorimotor function in patients with cervical cord disease. In Case III, the peg test score was 14.9 in stage B, and the gamma wave power values of Fz and Pz were attenuated. The peg test cutoff score for individuals with cervical SCI was set at 11 points [22], and the score of Case III exceeded the cutoff score. Motor learning is believed to minimize the cognitive control required for sensorimotor information processing, resulting in a decrease in the amplitude of motor-related cortical potentials in the sensorimotor area [44,45]. In Case III, Yubireco’s intervention improved dexterity, indicating that the cognitive control required for sensorimotor adaptation during dexterity was efficiently performed. The tactile discrimination compensation real-time feedback device used in this study enabled interventions tailored to the state of individual sensorimotor functions and contributed to the promotion of sensorimotor integration.

Regarding the sensory function evaluation, Case II showed moderate changes in Tau-U in the numbness NRS, and Case III showed very large changes in Phase B. Efficient sensorimotor processing during movement can block unnecessary sensory inputs [46]. This suggests that real-time tactile discrimination compensation feedback during dexterous movements blocked unnecessary sensory information during movement execution, thereby improving the numbness in Cases II and III. This result suggests that the device used in this study may have promoted the learning of appropriate motor outputs based on sensory input and contributed to the improvement of sensorimotor integration.

Regarding the learning assessment in MAL, all items in Cases I, II, and III showed a tendency towards improvement in B. In addition, regarding SoA, all cases showed a tendency for improvement in B. Taub et al. [26] investigated the effects of CI therapy using MAL in patients with upper limb motor dysfunction after stroke. CI therapy has been reported to significantly improve MAL scores and the use of the upper limbs in daily life. Thus, the effect of rehabilitation can be evaluated at the daily life level using the MAL. Gamma waves in the sensorimotor area reflect whether there is a discrepancy between motor intention and sensory feedback, regardless of the sensorimotor function severity [47]. The SoA is enhanced when sensory information appropriate for motor intentions is input in real time [48,49]. Motor learning has been reported to be facilitated owing to an increased SoA [49]. As the SoA increased in all cases, the intervention using Yubireco might have promoted the motor learning required for skilled movements, which might have improved the ability to perform living activities using fingers and the quality of finger use.

The improvement in the ability to perform daily activities using the fingers and the quality of finger use in all cases suggests that real-time tactile discrimination compensation feedback may be appropriate for patients with sensorimotor dysfunction of the fingers after cervical SCI. Recent studies have reported on the effectiveness of sensory feedback in individuals with hand sensorimotor dysfunction. Improvements in hand sensorimotor function and neuroplastic changes in the sensorimotor region of the cortex have been reported in post-stroke patients with hand sensorimotor dysfunction by sensing tactile information and providing compensatory feedback of tactile information [50]. This prior study suggested that interventions using sensory feedback associated with finger movements might be effective for treating finger sensorimotor dysfunction. It was suggested that this study is similar to previous studies. Furthermore, the results of gamma wave power analysis in the sensorimotor area suggest that different sensorimotor integration processes may occur during the recovery of dexterity in non-skilled and skilled participants. The novelty of this study is that it revealed differences in the neural mechanisms of sensorimotor integration depending on the severity of sensorimotor dysfunction in the fingers after cervical SCI, suggesting that selecting an appropriate rehabilitation method depending on the severity is important.

### 4.1. Limitations

The long-term impact of using this system cannot be determined from the results of this study alone. The possibility of chance cannot be ruled out because of the small number of participants in the intervention using this system. It is necessary to show the results statistically through a large-scale survey.

### 4.2. Future Challenges

In Case I, who had a low NRS score for finger numbness, the intervention using this system was ineffective for numbness. In the future, it will be necessary to increase the number of cases and conduct statistical verification to determine whether interventions using this system are effective in cases of mild finger numbness. The extent to which an increase or decrease in the neural activity of gamma waves in the sensorimotor region is clinically significant remains unclear. In the future, we will increase the number of cases and determine the minimal clinically important difference [51]. In addition, we will test the interventions using this system by changing the target disease and body parts. In addition to the questionnaire, it was necessary to determine whether the sensory feedback information appropriately matched the execution of the movement in order to determine whether a sense of agency was generated. To determine whether a correlation exists, the event-related potential will be measured using error-related negativity [52].

The novelty of Yubireco differs from the methods used in previous studies in that it feeds the tactile information associated with finger movements back to other body parts in the form of real-time vibrational stimuli. Therefore, real-time tactile discrimination compensation feedback using Yubireco may be more effective for finger sensorimotor dysfunction following cervical SCI.

In the future, it will be necessary to examine long-term effects and generalize these findings to daily life activities. It is also necessary to compare the intervention method used in this study with existing rehabilitation methods and examine its superiority. Based on the results of this study, appropriate rehabilitation programs should be established according to the severity of the condition.

These three case studies are a pilot study to examine the feasibility and safety of this novel intervention.

## 5. Conclusions

This study examined the effectiveness of an intervention using a tactile discrimination compensation real-time feedback device in patients with central nervous system afferent pathway injuries (sensory conduction pathways from sensory receptors to the cerebral cortex) presenting with finger sensorimotor dysfunction. As a result of this intervention, moderate or very large changes in numbness were observed in two of the three patients in stage B. Large or very large changes were observed using the peg test in all of the patients in stage B. Regarding EEG activity, an increase in gamma wave power in the sensorimotor area was observed in the non-skilled participants in stage B. Attenuation of gamma-wave power in the sensorimotor area during dexterous movements was observed in motor learning. These results indicate that the comparison and matching function between predictive control and sensory feedback in the brain may be impaired in patients with central nervous system afferents after cervical SCI and that intervention using this device contributed to the recovery of this function. Furthermore, our results suggest that different neural mechanisms of sensorimotor integration are involved in the recovery processes of nonskilled and skilled participants. The results of this study are expected to contribute to the development of rehabilitation strategies for sensorimotor dysfunction of the fingers after cervical SCI.

The primary novelty of this study is that it used a tactile perception discrimination compensation real-time feedback system for finger sensorimotor dysfunction following cervical SCI, which was verified using high-frequency EEG bands according to severity. We believe that this will contribute to the selection of appropriate treatment methods according to the severity of hand sensorimotor dysfunction.

## Figures and Tables

**Figure 1 brainsci-15-00352-f001:**
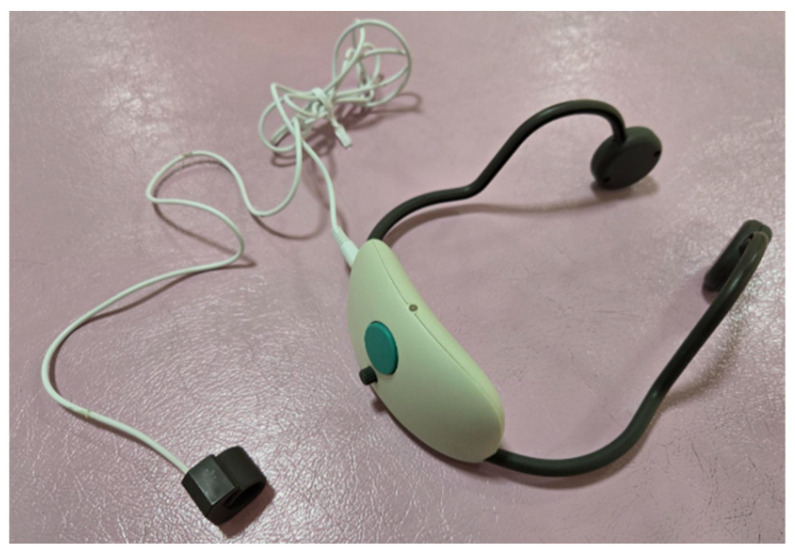
Finger recorder.

**Figure 2 brainsci-15-00352-f002:**
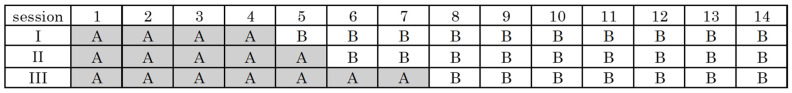
Multiple baseline design schedules. I: Case I, II: Case II, III: Case III.

**Table 1 brainsci-15-00352-t001:** Basic information on all cases and results of sensorimotor function assessment.

Cases	I	II	III
Gender, Age (yr)	Woman, 83	Man, 51	Man, 53
Height (cm), weight (kg), Body Mass Index	154.0, 42.6, 18.0	174.0, 55.5, 18.3	162.0, 59.1, 22.5
Diagnosis	Cervical spondylotic myelopathy (C3/4)	Cervical spondylotic myelopathy (C3–6)	Cervical spinal cord injury (C7)
Operative History	Cervical laminectomy (C3/4)	Cervical laminoplasty (C3–6)	None
Numbness Numerical Rating Scale	Right fingers 1	Right fingers 8	Right fingers 4
Purdue Pegboard Test (Quantity)	6.0	1.5	10.0
The American Spinal Injury Association score			
Motor Function	C6: 3/5	C6: 2/5	C6: 4/5
	C7: 3/5	C7: 2/5	C7: 5/5
	C8: 3/5	C8: 3/5	C8: 4/5
Sensory Function	C6: 2/3	C6: 1/3	C6: 1/3
	C7: 2/3	C7: 1/3	C7: 1/3
	C8: 2/3	C8: 1/3	C8: 1/3
Study start date	X+about 2 months	X+about 2 weeks	X+about 3 months
History	Hypertension, chronic gastritis, osteoporosis	None	Hypertension, insomnia

**Table 2 brainsci-15-00352-t002:** Electroencephalography results.

Gamma Wave (40–100 Hz)				
	Fz (dB)		Pz (dB)	
	A AVG ± SE	B AVG ± SE	A AVG ± SE	B AVG ± SE
I	0.015 ± 0.008	0.016 ± 0.019	0.008 ± 0.001	0.014 ± 0.013
II	0.074 ± 0.038	0.090 ± 0.036	0.007 ± 0.003	0.055 ± 0.023
III	0.744 ± 0.281	0.010 ± 0.003	0.852 ± 0.322	0.033 ± 0.017

dB: decibel, AVG: average, SE: standard error, I: Case I, II: Case II, III: Case III.

**Table 3 brainsci-15-00352-t003:** Purdue Pegboard results.

	A AVG ± SE	B AVG ± SE	Tau-U
I	6.50 ± 0.35	7.30 ± 0.26	0.67
II	2.30 ± 0.34	3.28 ± 0.30	0.67
III	11.9 ± 0.38	14.9 ± 0.26	1.00
Combined			0.80

AVG: average, SE: standard error, I: Case I, II: Case II, III: Case III, Combined: Combined cases I–III.

**Table 4 brainsci-15-00352-t004:** Numbness Numerical rating scale results.

	A AVG ± SE	B AVG ± SE	Tau-U
I	1.00 ± 0.00	1.00 ± 0.00	0.00
II	7.80 ± 0.20	7.11 ± 0.31	0.52
III	3.70 ± 0.49	1.00 ± 0.58	1.00
Combined			0.65

AVG: average, SE: standard error, I: Case I, II: Case II, III: Case III, Combined: Combined cases I–III.

**Table 5 brainsci-15-00352-t005:** Motor activity Log-14 results.

I	AoU	QoM	SoA
Initial	2.85	2.46	5.69
A	3.08	2.54	5.69
B	3.62	3.23	6.92
II	AoU	QoM	SoA
Initial	0.64	0.43	0.79
A	1.14	1.36	1.29
B	1.50	1.43	1.64
III	AoU	QoM	SoA
Initial	4.07	4.07	8.29
A	4.21	4.21	8.71
B	4.71	4.71	9.43

AoU: amount of use, QoM: quality of movement, SoA: sense of agency, I: Case I, II: Case II, III: Case III.

## Data Availability

Data supporting the results of this study are available from the corresponding author, T.K., upon reasonable request.

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
