# Peer review of "Enhancing Hand Sensorimotor Function in Individuals with Cervical Spinal Cord Injury: A Novel Tactile Discrimination Feedback Approach Using a Multiple-Baseline Design"

_brainsci, 2025, doi:10.3390/brainsci15040352_

Round 1
Reviewer 1 Report
Comments and Suggestions for Authors
Thank you for submitting this interesting paper describing the three case studies. I particularly like to encourage publication of case studies as I feel that they are more relevant to the neurological patient population. I also comment you on the use of the TAU-U for analysing your results.
I have indicate don the manuscript where I feel changes are required. I would however ask that the materials and methods section be rewritten as I found it hard to interpret what was being said. As I have indicated, a photograph or schematic diagram of the participant set-up could greatly enhance the clarity of this section.

Comments on the Quality of English Language
I have commented in the manuscript where there are English language problems.
Reviewer 2 Report
Comments and Suggestions for Authors
Dear authors, the document you submitted, entitled "Enhancing hand sensorimotor function in individuals with cervical spinal cord injury: A novel tactile discrimination feedback approach using a multiple-baseline design" is very interesting and nicely written. Some suggestions are made to improve the overall quality of the document.
In the introduction section, lines 48-50, please make it clear the differences are between conservative and surgical treatments, since it is not clear for the reader. Furthermore, at the end of the introduction section, the last paragraph, should clearly state the aim and objectives of the present study, I would suggest you rephrase the last paragraph.
In your methods section, line 227-229, please justify the use of a bandpass filter of 4-30Hz to study gamma waves (40-100Hz). Furthermore, it is unclear why you used different amplifications for each participant (lines 289-294).
In your results section, please include a legend describing each acronym after each table.
At the end of your discussion section, please include a list of limitations from your study.
Kind regards
Reviewer 3 Report
Comments and Suggestions for Authors
Dear authors,
I have embedded my comments in the actual text. The overall manuscript needs to be rewritten and needs to provide clear figure. Your should have the phrase "case series" in your manuscript title. Your work is not any different from previous case reports in the field especially with only three participants. The characteristics of the three participants are missing, body weight, height, BMI. You have only included age. You also need to include a clear figure highlighting the actual device that used in the study and how it works. The way is currently described is very difficult to understand or to follow your overall application. Overall, the entire paper needs to be significantly shorten at different sections.

Comments on the Quality of English Language
The paper needs extensive English language editing and needs to be proofread by a native english speakers. Several sentences lack grammatical structure and does not make sense.
Reviewer 4 Report
Comments and Suggestions for Authors
The manuscript “Enhancing hand sensorimotor function ….” examines the effects of using a device compensating for loss of hand sensorimotor disfunction in patients with an incomplete spinal cord injury. In general, there are many fragments which are hard to grasp immediately due to absence of initial definitions of the technique details or working terminology, which can appear later in the text or can be deduced from the following context.
Abstract
- 20 -- Define what baseline period and intervention periods are; or better do not use AB notion before it is defined in the main text.
- 32 -- “… the brain may be impaired in patients with CNS afferent pathways, ….”
What does it mean? All people have CNS afferent pathways regardless of their health.
Introduction
- 64 -- since you refer to this line later in #85, you should specify the type of stimulation, (apparently it is electrical stimulation) or remove “aforementioned” from #85.
- 85-87 -- “…electrical stimulation, which aims to increase the number of firings in the spinal tract and spinal anterior horn, has a different movement pattern …??? What is it? Which exactly spinal tract (ascending descending, thalamo-cortical etc.) do you mean? First, neither a spinal tract nor the anterior horn can fire! It is the firing of spinal neurons that can be induced or increased by electrical stimulation, so trains of their action potentials propagate through their axons in corresponding spinal tracts. Second, the electrical stimulation cannot have a movement pattern. Please, reformulate.
- 92-94 -- “.. converts … into vibration information, and provides compensatory feedback to other body parts ..” How? In what form is this feedback provided? Amplitude and/or frequency modulated vibration delivered to some body part?
- 95 -- “ … sensorimotor region “ of the cortex …
- 96-100 -- Does it mean low-frequency EEG (alpha-theta-beta?) parameters were converted into vibration? Be clear, say it directly.
- 103 -- “However, all these reports …” There are only two reports mentioned in this paragraph: [16] and [17]. Therefore, better “However, both [16] and [17] reports…”.
- 112 -- “..the high-frequency band of the EEG in the sensorimotor area …”
- 125-126 -- “.. verified using high-frequency EEG bands according to severity,…” Does the severity of disfunction correlate with the power of HF in EEG? If so, please, make a reference.
Methods
- 183-186 -- Describe here in detail the AB design: what is the baseline period (is it the control period with tests before treatment?) and what is the intervention period (is it just treatment period with tests paired with the feedback?). Make it clear here.
- Figure 1 -- a) If A and B designate the baseline and interventions in the text, why in the table BL and IV are used? Use A and B in the table for consistency; b) Do I, II and III designate patients? Indicate in in the legend; c) Explain in the legend or in text the reason why durations of BL=A (and IV=B) are different for patients.
- 209-213 -- It seems that most of this paragraph describes period A, and in #214 you describe period B. Indicate it directly, so that in B the finger recorder is added with all the rest being the same. Thus, A serves as a control for the following treatment (i.e. for B).
- 217 -- Longitudinal …
- 218-219 -- ..” before Intervention A started,…” If A is an Intervention, it cannot be a baseline (as defined previously). Clarify as early as possible in the text what A really stands for. It should be no ambiguity.
- 265 -- Please, name the independent components (parameters?) extracted from the EEG. Do they differ from conventional parameters like average amplitude of EEG waves, power of spectral peaks etc.?
- 271-272 -- Since the bandpass filter for EEG recording was set at 4-30 Hz (see # 228), the gamma wave band (40-100 Hz) must be either filtered out or its power must be significantly weakened. Explain why you filtered EEG for theta/beta waves, whereas you analyzed the gamma range of EEG.
- Fz and Pz recording points correspond to the frontal and visual cortex areas, but not to the sensorimotor area. Why were these recording points selected while you studied sensorimotor functions?
Results
- 290-294 -- What do the numbers mean? Are they spectral powers in the gamma range? Please, define. What are the units for these measurements?
- “Two-thirds of the patients…” It is not a correct definition in the case of only three available patients. This could be said if you collect data from 30 patients, but not from 3.
Discussion
- Were EEG recordings from points other than Fz and Pz analyzed? Provide the foundation or references for Pz and Fz selection in your work.
Round 2
Reviewer 1 Report
Comments and Suggestions for Authors
Thank you for adjusting your manuscript according to the reviewers comments.
It is reading much better now, but there are still some changes needed. These I have indicated in the manuscript itself.

Comments on the Quality of English Language
I have commented throughout on the English and have offered alternatives in most cases.
Reviewer 2 Report
Comments and Suggestions for Authors
Dear authors, thank you for submitting the revised version of the document. It is now more clear.
At the end of your introductory chapter please state the aims and objectives of your study clearly. Example: the aim of the present study is to....
In your methods section, please describe the individual filtered frequencies, because the reader should be able to clearly see how the signal was treated. Please explain, in the document, the need for different signal amplification.
In your point 6, please consider moving the limitations of your study to the end of the discussion section and the directions for future studies to the end of the conclusions. This way, this point 6 should disappear.
Kind regards
Reviewer 3 Report
Comments and Suggestions for Authors
The authors have addressed most of my previous comments.
Comments on the Quality of English Language
The paper still needs to be proofread for English language. Several sentences throughout the manuscript needs structural and grammatical corrections.
Reviewer 4 Report
Comments and Suggestions for Authors
Thank you for considering my comments. The revised manuscript is now much easier to comprehend for readers.
However, the subtitle on line #488 should start from the capital letter:
6. limitations #488
The long-term impact of using this system cannot be determined from the results of #489
Round 3
Reviewer 2 Report
Comments and Suggestions for Authors
Dear authors, thank you for submitting the revised version of your manuscript. Some suggestions are made to improve the overall quality of the document.
The aim of the study should be clearly stated and consistent throughout the document. It should include the population, the intervention, and the outcome measures. Please rewrite your aims and objectives for clarity.
Kind regards
